# Economic policy uncertainty, intra-industry trade, and China's mechanical and electrical product exports

Dajun Liu[1], Xiugang Zhu [2]*, Huiru Yu [3]

1 School of Economics and Trade, Anhui Finance & Trade Vocational College, Hefei, China, 2 Bengbu Finance Bureau, Bengbu, China, 3 School of Economics, Hefei University of Technology, Hefei, China

* 2393686732@qq.com, 18756589046@163.com

## Abstract

Economic policy uncertainty has had an important impact on trade and sustainable economic development. Especially in some specific industries, uncertainty has increased dramatically. The extant related literature mainly analyzes the nexus between uncertainty and trade across different industries and focuses less on a specific industry. Using Chinese customs data on HS 8-digit products over the period of 2000–2013, this paper first investigates the impact of both foreign economic policy uncertainty (EPU) and domestic intra-industry trade on China's mechanical and electrical product exports to 23 trading partners and applies pooled OLS regressions to conduct an empirical study. This paper finds that EPU has a significant inhibition effect on mechanical and electrical product exports; conversely, intra-industry trade can both significantly promote exports and alleviate the inhibition effect of EPU. In addition, the export impact of EPU varied with different trade patterns. It can significantly inhibit processing exports, while it has no effect on ordinary exports. The results of this paper indicate that in the context of increasing uncertainty, our findings could have far-reaching policy implications for China to build a new development pattern of domestic and international dual circulation.

## 1. Introduction

Recent years have witnessed an increase in global economic uncertainty and the rise of trade protectionism, and frequent adjustments of economic policies in foreign countries have increased obstacles for the smooth development of China's trade. The academic community has extensively analyzed the relationship between uncertainty and exports [1] and has further discussed the stability and resilience of the supply chain [2]. Existing literature has mainly concentrated on the trading figures of developed countries and has revealed two mechanisms through which EPU may take effect: the option value of waiting and the investment portfolio. Based on the investment irreversibility and real option theory proposed by Bernanke (1983) [3] and Dixit (1989) [4], one side argues that uncertainty has a negative impact on trade flows and welfare by reducing firms' current exports [5, 6]. Real options theory is a modern theory on how to make decisions about investments when the future is uncertain. When the

**Data Availability Statement:** The data is available on http://www.policyuncertainty.com/; http://www.cepii.fr/cepii/en/bdd_modele/bdd.asp; and http://tariffdata.wto.org/reportersandproducts.aspx.

**Competing interests:** The authors have declared that no competing interests exist.

uncertainty risk is large, investors who choose instant investment can obtain instant investment profits but lose future possible profits and may also bear losses; delayed investment loses instant investment profits but may gain future profits. Another possibility is that the uncertainty is further expanded, so the loss outweighs the gain. The other side thinks that firms may actively diversify their trade risks through Markowitz portfolio strategies [7]. Markowitz portfolio strategies were proposed in 1952, which holds that investors' investment desire is to pursue high expected returns and avoid risk as much as possible. Based on the established risk level, how to make the possible expected return of securities great, or to obtain the established expected return, how to make the risk minimal.

The extant related literature mainly analyzes the nexus between uncertainty and trade across different industries and focuses less on a specific industry. Relying on the comparative advantage of its labor force, China has made great efforts to develop its intra-industry trade and has gradually embedded itself into the global value chain through processing trade with a feature of "two ends abroad." Although such a pattern of trade can play a role in attracting foreign investment and promoting domestic employment, it may also lead to some uncertainty in exports: On the one hand, since the product supply chain is distributed globally and the final products generally need to be exported, processing trade is more easily affected by unstable factors in international trade; on the other hand, processing trade is mainly dominated by multinational firms with a strong ability to obtain and distinguish information, and with relatively lower fixed costs, processing trade can avoid, to some extent, the influence of international market uncertainty. Therefore, to analyze the impact of EPU on China's exports, we need to consider the important role of firms' trade patterns.

Extensive literature on the relationship between EPU and trade [6, 7] does not consider the correlation between the characteristics of intra-industry trade and EPU. The impact of intra-industry trade may be more sensitive to EPU. At the same time, the literature focuses less on the impact of a certain industry and ignores the differences in the impact of uncertainty on the trade of different industries. In this context, the paper focuses on the electromechanical products industry to address this research gap. This paper aims to clarify the impact mechanism of trade uncertainty on export trade within specific industries. On one hand, it will facilitate our understanding of the changing trends in foreign trade under the backdrop of uncertainty from a global value chain perspective. On the other hand, it will be beneficial in providing relevant policy insights for emerging market countries like China to address economic policy uncertainties. This paper focuses on the impact of EPU on exports of the mechanical and electrical products industry and conducts an empirical study using customs data. We find that EPU has a significant inhibition effect on mechanical and electrical product exports; conversely, intraindustry trade can both significantly promote exports and alleviate the inhibition effect of EPU. In addition, the export impact of EPU varied with different trade patterns. It can significantly inhibit processing exports, while it has no effect on ordinary exports.

The mechanical and electrical products mentioned in the article refer to the products whose HS code range from 82 to 91 in the customs data. We set our research object as China's exports of mechanical and electrical products for three reasons. First, China is the largest trading country in the world, and so it is representative. Mechanical and electrical products, as the guarantee of China's export growth, account for the largest proportion of China's exports. China's mechanical and electrical products are constrained by raw materials and energy. When the supply is short, the fixed costs of enterprises will rise. Therefore, the exports of the mechanical and electrical industries are more sensitive to changes in fixed costs. According to the theory, the mechanical and electrical industries should be more affected by EPU. As shown in Fig 1, the proportion of exports of electrical and mechanical products to total exports has been above 50% since 2003. Second, the exports of such types of products can easily be targeted

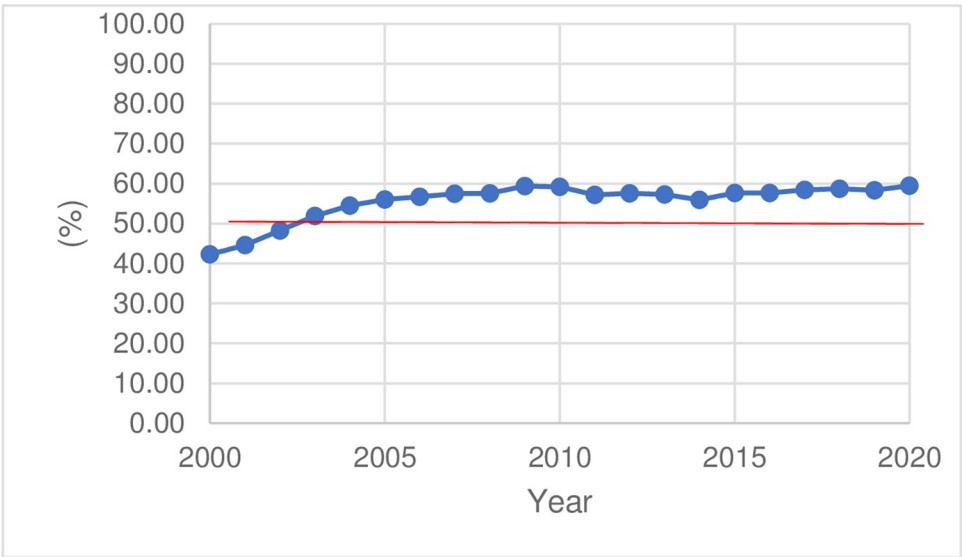

**Fig 1. The proportion of the export volume of mechanical and electrical products.**

directly by tariffs and other trade policies. Judging from the initial tariff list of the US after the Sino-US trade friction, the proportion of taxation on mechanical and electrical products is as high as 74.33%. Third, considering that the production of mechanical and electrical products involves hundreds of parts and components, the phenomenon of global production and sales is more pronounced and has a typical feature of a high proportion of processing and intra-industry trade.

The core issue of this paper is to identify the impact of EPU on mechanical and electrical products and the moderating role of intra-industry trade. To make the empirical results more credible, we addressed the endogeneity problem in three aspects. First, the omitted variable generally causes an error term correlated with other explanatory variables, which leads to a violation of the exogenous assumption. To solve this problem, we further control the product-country fixed effects according to the baseline regression. Second, the presence of measurement error biases causes explanatory variables to be correlated with perturbation terms, resulting in an endogeneity problem, so we use the weighted arithmetic mean method to recalculate the EPU index. Finally, reverse causality also causes a potential endogeneity problem, and we lag the intra-industry trade index for one period as a proxy to address this problem. After alleviating the three endogeneity problems, the conclusions remain robust.

The marginal contribution of this paper is mainly reflected in three aspects. First, the nexus between uncertainty and exports is analyzed from the perspective of a specific industry. The extant related literature mainly analyzes the nexus between uncertainty and trade across different industries and focuses less on a specific industry. Second, we empirically test the moderation effect of intra-industry trade and trade patterns on the relationship between uncertainty and exports based on highly disaggregated trade data. Third, the conclusions have some crucial policy implications. Further involvement in intra-industry trade and diversification of trade patterns can help firms alleviate the impact of external shocks due to policy uncertainty.

The remainder of the paper is structured as follows. We show the relevant literature and theoretical hypotheses in Section 2. Based on the related literature, we propose three theoretical hypotheses to be tested. We present the estimation strategy in Section 3, mainly by establishing the regression models. Section 4 shows the regression results, including the benchmark

regression results and a series of robustness tests. Section 5 presents the conclusion and policy implications.

## 2. Related literature and theoretical hypotheses

### 2.1. EPU and exports

The linkage between EPU and macroeconomic or microfirm dynamics has always been a hot topic in the trade literature. Recent studies tend to believe that the rise of EPU can increase the volatility of macroeconomic variables and thus has an inhibitory effect on the macroeconomy [8–13], while in terms of microfirm dynamics, related studies have shown mixed results. Some find that EPU, as an external risk faced by firms, can increase firms' option value of waiting and make them adopt prudent management strategies such as delaying investment, thereby increasing firms' financing constraints and precautionary cash holding and inhibiting firms' investment [14]. Shepotylo and Stuckatz (2017) [15] focus on manufacturing enterprises in Ukraine and find that trade policy uncertainty (TPU) can significantly inhibit FDI inflows. Kim and Nguyen (2018) [16] use data of 881 firms from eight East Asian countries between 2003–2013 and find that multinational corporations tend to invest in countries with lower uncertainty. With the improvement of EPU measurement methods, a great deal of relevant literature has emerged in recent years, especially on the links between EPU and trade. Based on the option value of waiting theory, relevant studies find that external EPU mainly inhibits the extensive margin of exports. For example, Greenland et al. (2014) [17] document that EPU can pronouncedly suppress the extensive margin of exports but has no effect on the intensive margin. Similar conclusions have been found in different studies on different countries, such as Handley (2014) [18] on Australia, Handley and Limo (2015) [19] on Portugal, Limo and Maggi (2015) [20] on the US and Cuba, and Osnago et al. (2018) [21] on 149 countries. Moreover, Greenland et al. (2019) [6] also documented the significantly indirect promotion effect of EPU on the intensive margin of exports.

Previous studies have provided strong theoretical and empirical evidence for our work. Compared with ordinary manufacturing products, mechanical and electrical products have the features of high technology, rapid technology renewal, and large trade value, so they have larger fixed export costs and are more sensitive to changes in fixed costs. The increase in EPU can raise the cutoff productivity of exporting firms, resulting in massive mechanical and electrical product exporters with lower productivity cutting back their current value of exports or even withdrawing from the export market [15, 21], thereby reducing the total export scale. Since mechanical and electrical products have the features of high durability and low demand stickiness and exports need to meet various technical regulations, higher EPU, on the one hand, increases the policy regulation risks of mechanical and electrical product exports. On the other hand, it also reduces foreign demand, thus reducing firms' exports for risk avoidance.

**Hypothesis 1 (H1)**. EPU can significantly suppress China's exports of mechanical and electrical products.

### 2.2. Intra-industry trade and exports

Intra-industry trade, as defined by Balassa, is the simultaneous import and export of the same category of goods in a certain industry. Intra- and interindustry trade, as the two main patterns of international trade, are mainly focused on in the classical trade literature. In terms of the economic impact of intra-industry trade, the current literature has shown mixed findings. Li's investigation of Asian countries (2017) [22] suggests that intra-industry trade can significantly

increase the synchronization degree of the business cycle. Hoang (2019) [23] focuses on the agricultural industry in Vietnam and finds that intra-industry trade plays an important role in international trade but hinders the improvement of trade specialization. Using panel data between Malaysia and its major trading partners from 1988 to 2016, Chin et al. (2020) [24] find that intra-industry trade can significantly increase the national economic scale. Neumann and Tabrizy (2021) [25] studied the impact of the exchange rate on imports, exports, and trade balance at the industry level, relying on data from 13 manufacturing industries in five Asian countries from 2001 to 2015. The study finds that vertical specialization can reduce the exchange rate elasticity of trade, while intra-industry trade has an adverse effect. Dimopoulous et al. (2011) [26] on Mexico, Zhang et al. (2013) [27] and Feng (2018) [28] on China all find that intra-industry trade has a significant negative impact on the skill premium in manufacturing, while Liu and Wei (2012) [29] and Ding and Yu (2014) [30] have adverse findings from the former.

In terms of the export impact of intra-industry trade, the literature mainly focuses on how exports affect industrial trade. Yoshida (2013) [31] uses Japan-South Korea trade data from 1998 to 2006 and finds that the increase in the export categories promotes the development of intra-industry trade, while the increase in the existing value of exports has a converse effect. In general, intra-industry trade is more likely to occur between countries or regions with similar industrial structures and may affect exports in two ways. On the one hand, highly developed intra-industry trade may lead to homogeneous competition among similar countries, thus restraining exports; on the other hand, intra-industry trade can closely link the domestic and international circulations of production in a country, which helps firms make use of both domestic and foreign resources, thus promoting a country's exports. As a typical capital- and technology-intensive industry, the mechanical and electrical industry, with a corresponding index above 60, is more likely to experience intra-industry trade. Considering the coexistence of very large trade value and intra-industry trade, we hypothesize that the promoting effect of intra-industry trade is greater than the inhibiting effect of product homogenization.

**Hypothesis 2 (H2)**. Intra-industry trade can significantly promote China's exports of mechanical and electrical products.

## 2.3. The moderating effects of intra-industry trade

The export impact of EPU on mechanical and electrical products varies with the degree of intra-industry trade. This can be explained from three aspects: information asymmetry, scale economy, and demand stickiness. Classical trade theories generally assume perfect information in trade. However, access to information in actual life is not completely free, and the completeness of information depends, to a certain extent, on the ability to pay for information acquisition costs. Information asymmetry, as a normal phenomenon in real life, is now a great barrier in international trade. With EPU increasing in the export market, the difficulty of obtaining product information between trading partners, as well as the technical barriers to trade (TBT) and trade friction risk, all become greater, which may result in additional losses. For industries with higher levels of intra-industry trade, the costs of information acquisition and adjustment are relatively lower; thus, the degree of information asymmetry is also lower. Different from interindustry trade featuring comparative advantage, the characteristics of intra-industry trade are scale economy and differentiated products. Since the technology spillover between differentiated products within the same industry is easier, intra-industry trade deepens the specialization among trading partners, thus forming economies of scale and reducing production costs. In addition, the more frequently the imports and exports of

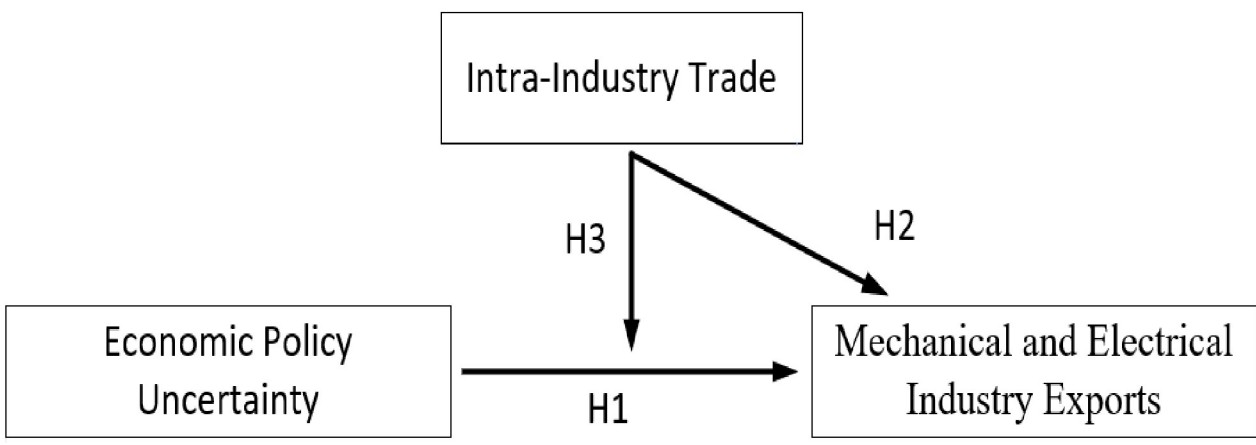

**Fig 2. Theoretical mechanism.**

differentiated products take place between the two countries, the greater the difference in consumer preferences and the stronger the stickiness of demand, thus effectively resisting the impact of EPU.

**Hypothesis 3 (H3).** Intra-industry trade can alleviate the negative export impact of EPU on mechanical and electrical products.

The classic theories relevant to this paper are primarily the theory of monopolistic competition in international trade [32, 33]. This theory is mainly based on empirical facts from developed countries and explains the role of internal economies of scale and firm heterogeneity in international trade patterns. Subsequently, some scholars incorporated uncertainty into the analytical framework and extended the models of heterogeneous firm trade from various perspectives [1, 5]. The theoretical framework of this article primarily highlights the significant role of economic policy uncertainty in global trade, particularly when it comes to specific industries, where uncertainty can sharply increase. Within this theoretical framework, the article explores the relevant research questions, and the theoretical mechanism is illustrated in Fig 2. Fig 2 is the mechanism diagram depicted in this paper, where H1 illustrates the inhibitory effect of EPU on the export of mechanical and electrical products, H2 indicates the promotion effect of intra-industry trade on the export of mechanical and electrical products, and H3 indicates that it can alleviate the inhibitory effect of EPU on the export of mechanical and electrical products.

## 3. Estimation strategy

### 3.1. Data

Our empirical analysis relied heavily on the panel data of mechanical and electrical products from the Chinese customs database between 2000 and 2013, including highly detailed export information at the firm-export-location-destination-product-pattern level. Although the latest version commonly used in academic circles was only updated in 2013, the two periods of the rapid rise of global EPU in 2001 and 2008 could provide an effective economic background for our study. Compared with the sample data of Tian et al. (2020) [34], we further included the samples with export destinations of Hong Kong and Croatia, which is the most complete range available for an EPU index at present. In 2013, China's export value of mechanical and electrical products to the sample countries (regions) accounted for 76.9% of the total, supporting the strong representativeness of our samples.

### 3.2. Variable

**3.2.1. Explained variables.**   Our explained variables included the export value (*lnvoh*) and ordinary export value (*lnvoh_ot*) of mechanical and electrical products, as well as the processing exports' value with imported materials (*lnvoh_jlpt*) and foreign supplied materials (*lnvoh_llpt*). Moreover, to explore the mechanism through which the export impact of EPU is verified from different trade patterns, we further divided the total value into quantity and price margins and obtained export price (*lnprice*) and export quantity (*lnqoh*).

**3.2.2. Key explanatory variables.**   We directly applied the EPU index (lnEPU) published by Baker et al. (2016) [35] as the measure of our key explanatory variable and used the simple arithmetic mean method to convert the original monthly data into the annual level. We further used the weighted arithmetic mean method to recalculate the EPU index (lnWEPU) for robustness checks.

**3.2.3. Moderating variable.**   We used the Grubel–Lloyd (GL) index (1975) [36] and Brulhart marginal intra-industry trade index (1994) [37] to identify the moderating effects. The GL index is by far the most authoritative intra-industry trade index. The main idea is that the larger the share is, the deeper the degree; thus, the calculation formula is as follows:

$$GL_k = \left[ 1 - \frac{|X_k - M_k|}{X_k + M_k} \right] \times 100$$

where we use the subscript k to distinguish different industries; thus, $GL_k$ represents the horizontal intra-industry trade index of China's industry k, and $X_k$ and $M_k$ represent the export value and import value of industry k, respectively. $GL_k$ reflects the proportion of overlapping trade occurring in intra-industry or intraproduct trade, and its value is between 0 and 100. The larger the value of $GL_k$, the higher the degree of intra-industry trade, and 0 and 100 were taken for complete intra-industry trade and complete intra-industry trade, respectively. A GL index higher than 50 indicates an intraindustry-dominated pattern, while a GL index lower than 50 indicates an interindustry-dominated pattern. Although the GL index is widely used in academic circles, it cannot reflect the dynamic trend of trade patterns because of the absolute trade value. For this reason, Brülhart constructed a new intra-industry trade index from the perspective of value changes to dynamically measure the development trend of intra-industry trade and the degree of intrasector symmetry of trade changes. The new calculation formula is as follows:

$$Bi_k = \left[ 1 - \frac{|\Delta X_k - \Delta M_k|}{|\Delta X_k| + |\Delta M_k|} \right] \times 100$$

where $\Delta X_k$ and $\Delta M_k$ capture the increment of export value and import value of industry k, respectively, in a specific period. In the same way, we took 0 and 100 for complete interindustry trade and complete intra-industry trade, respectively. We used the GL index for baseline regressions and the Brülhart index for robustness checks.

**3.2.4. Control variable.**   Outside the EPU of the destination markets, we further considered other factors that may take effect, e.g., the economic scale of destination markets ($lngdp_{ct}$), free trade degrees ($lnfree_{ct}$), exchange rates ($lnxr_{ct}$), and tariffs ($tariff_{kct}$). Our empirical sample included 409,804 items of the product (HS 8-digit)-country-year unbalanced panel data from China and 23 trading partners. Table 1 reports the descriptive statistics and sources of all variables.

### 3.3. Estimation strategy

Our main work was to identify the export impact of EPU on mechanical and electrical products from the perspective of dual trade patterns, which included processing trade, ordinary

**Table 1. Descriptive statistics and sources.**

| Variable | Symbol | Mean | Std. dev. | Min | Median | Max |
|---|---|---|---|---|---|---|
| Explained variable | | | | | | |
| Export value | lnV | 12.58 | 2.97 | 3.91 | 12.70 | 24.95 |
| Export value (ordinary) | lnV_ot | 12.13 | 2.88 | 0.69 | 12.23 | 24.75 |
| Export value (processing) | lnV_pt | 12.27 | 2.91 | 0.00 | 12.38 | 23.64 |
| Export value (processing with imported materials) | lnV_jlpt | 11.93 | 2.92 | 0.69 | 12.01 | 23.50 |
| Export value (processing with imported materials) | lnV_llpt | 11.25 | 2.87 | 0.69 | 11.31 | 22.74 |
| Export price | lnP | 3.84 | 3.06 | 0.00 | 2.86 | 19.84 |
| Export quantity | lnQ | 9.02 | 4.46 | 0.00 | 9.53 | 25.14 |
| Key explanatory variable | | | | | | |
| Economic policy uncertainty of destination markets | lnepu_ave | 4.65 | 0.38 | 3.37 | 4.66 | 5.72 |
| Moderating variable | | | | | | |
| GL index | GL | 18.71 | 27.72 | 0.00 | 2.35 | 100.00 |
| Brülhart index | Bi | 12.32 | 24.29 | 0.00 | 0.00 | 100.00 |
| Control variable | | | | | | |
| Free trade degree | lnfree | 4.29 | 0.29 | 2.98 | 4.38 | 4.55 |
| Gross domestic product | lngdp | 27.56 | 1.27 | 24.27 | 27.69 | 30.45 |
| exchange rate | lnxr | -0.20 | 2.49 | -2.72 | -1.59 | 5.85 |
| Tariffs | tariff | 0.04 | 0.06 | 0.00 | 0.02 | 2.68 |

Source: The variables are calculated by the authors based on the mechanical and electrical product panel data from the China Customs database from 2000 to 2013 and the export information at the firm-export-location-destination-product-pattern level. See above for more information on variable constructions.

trade, interindustry trade and intra-industry trade. The former was tested in a subsample way, and the latter was tested in a moderating-variable way. To this end, we set up the following econometric model for empirical regressions.

To address the problem of omitted variables, we further considered a variety of fixed effects in our regressions. We included the country fixed effects for some consistent factors, such as bilateral distance, time fixed effects for some macroeconomic factors that varied only with time, and product fixed effects for some inherent characteristics at the product level.

We first set up regression Model (1) to identify the overall trade effects of EPU on mechanical and electrical products and then added the term of intra-industry trade index for the direct effects of intra-industry trade.

$$Y_{ict} = \beta_0 + \beta_1 EPU_{ct} + \beta Z + \lambda + \varepsilon_{ict} \tag{1}$$

Finally, we introduced the interaction term between EPU and intra-industry trade to test the moderating effect of intra-industry trade.

$$Y_{ict} = \alpha_0 + \alpha_1 EPU_{ct} \times GL_{kct} + \alpha_2 EPU_{ct} + \alpha_3 GL_{kct} + \alpha_4 Z + \lambda + \varepsilon_{ict} \tag{2}$$

where the subscript $i$ denotes the HS 8-digit product, $k$ refers to the HS 6-digit industry, $t$ for year, $c$ for country, $Y_{ict}$ represents the export value at the product-country-year level, $EPU_{ct}$ for the economic policy uncertainty index at the country level, $GL_{kct}$ for the intra-industry trade index at the HS 6-digit industry level, $EPU_{ct} \times GL_{kct}$ for the interaction term of EPU and intra-industry trade, Z represents a set of control variables, $\lambda$ for a set of fixed effects, and $\varepsilon_{ict}$ for the error term. Provided that the coefficient of interaction term $EPU_{ct} \times GL_{kct}$ shares the same sign with $EPU_{ct}$, intra-industry trade will intensify the export effect of EPU; otherwise, intra-industry trade will alleviate the export effect.

We use the increment of EPU and mechanical and electrical products to draw pictures, and as seen in Fig 3, there is a significant negative correlation between mechanical and electrical product exports and EPU, which is also consistent with the conclusions we draw.

## 4. Empirical analysis and findings

### 4.1. Baseline results

We first examined the linkage between EPU and the exports of mechanical and electrical products. The results are shown in Table 2. Column (1) considers only the key explanatory variable EPU, together with some control variables and time, country, and product fixed effects. The coefficient of our key explanatory variable lnEPU is negative and statistically significant, indicating that EPU in destination markets significantly suppresses the exports of mechanical and electrical products, which verifies Hypothesis 1. This conclusion is consistent with the findings of the literature [38, 39].

Based on Column (1), Column (2) additionally includes the intra-industry trade index, and the coefficient of lnEPU remains negative and statistically significant, while the coefficient of GL is positive and statistically significant, implying that intra-industry trade significantly promotes the exports of mechanical and electrical products, which verifies Hypothesis 2. We further included the key interaction term of intra-industry trade and EPU in Column (3), and the coefficient of lnEPU*GL is positive and statistically significant, suggesting that intra-industry trade can significantly alleviate the inhibitory effect of EPU on the exports of mechanical and electrical products, which verifies Hypothesis 3.

### 4.2. Robustness checks

To assess the strength of our results, we needed to address three aspects of endogeneity problems: omitted variables, measurement error, and reverse causality.

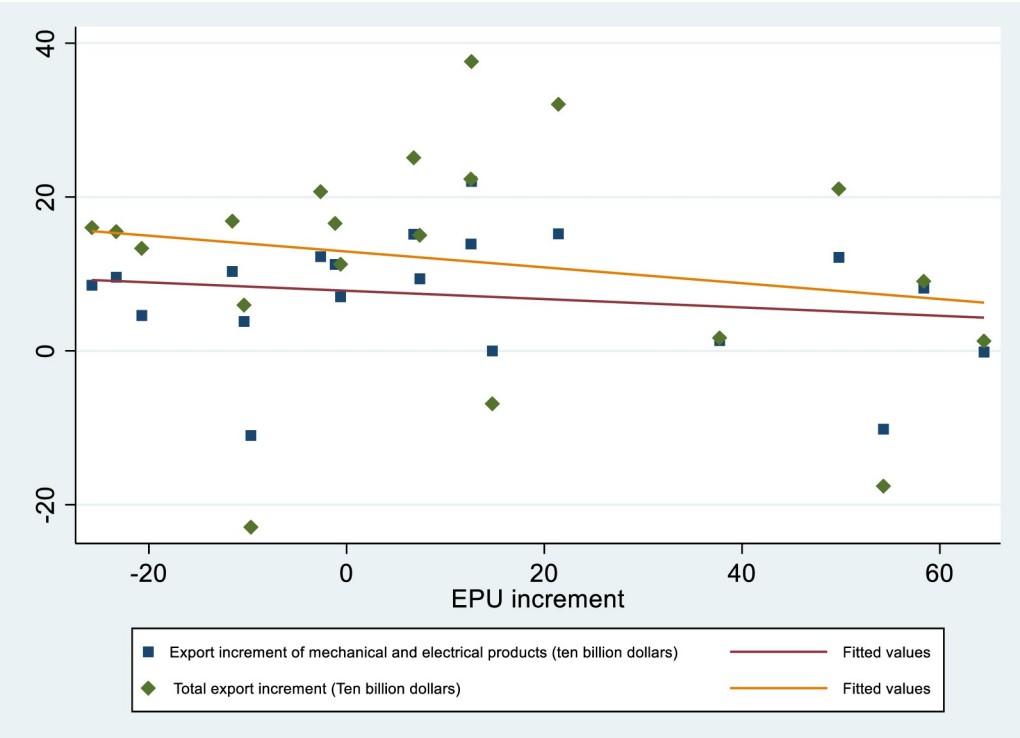

**Fig 3. Scatter plot of EPU increment and export increment.**

**Table 2. Baseline results.**

| Variable | (1) | (2) | (3) |
|---|---|---|---|
| | lnvoh | lnvoh | lnvoh |
| lnEPU | -0.1096*** | -0.0996*** | -0.0964*** |
| | (-5.5735) | (-5.0686) | (-4.9070) |
| GL | | 0.0041*** | 0.0040*** |
| | | (14.5981) | (14.3714) |
| lnEPU×GL | | | 0.0026*** |
| | | | (6.0467) |
| lnfree | 0.1161*** | 0.1162*** | 0.1132*** |
| | (4.3107) | (4.2997) | (4.1886) |
| lngdp | 1.2289*** | 1.2362*** | 1.2521*** |
| | (34.2696) | (34.4988) | (34.8526) |
| lnxr | 0.7911*** | 0.7988*** | 0.8073*** |
| | (12.6412) | (12.7487) | (12.8907) |
| tariff | -1.6905*** | -1.7351*** | -1.7429*** |
| | (-6.5051) | (-6.6578) | (-6.6895) |
| Constant | -21.0483*** | -21.8995*** | -21.3711*** |
| | (-21.0333) | (-21.9858) | (-21.3652) |
| Time FE | Yes | Yes | Yes |
| Country FE | Yes | Yes | Yes |
| Product FE | Yes | Yes | Yes |
| N | 408291 | 408291 | 408291 |
| adj. R2 | 0.459 | 0.460 | 0.460 |

Note:

\*\*\*, \*\*, and \* denote $p < 0.01$, $p < 0.05$, and $p < 0.1$, respectively. Columns (1)-(2) report the extent to which different variables are added to the connection between EPU and mechanical and electrical product exports. Column (3) shows the regression results for the interaction terms. The control variable exchange rate is measured by the exchange rate of RMB to foreign currency, and a greater exchange rate means a depreciation of RMB. The core explanatory variables and the regulatory variables were centralized when testing for the regulatory effects. FE = fixed effect.

First, we considered omitted variable biases. Although we considered country, product, and year fixed effects in our previous regressions, some omitted variables may still have existed. For example, the brand preference of specific countries for specific mechanical and electrical products (mobile phones, computers, etc.) may have taken effect. Therefore, we further considered the product-country fixed effects based on the baseline regressions, the results of which are reported in Columns (1)–(2) in Table 3, where the key interaction term of intra-industry trade and EPU is not included in Column (1), while in Column (2), it is included. After considering the more detailed fixed effects, EPU in destination markets still significantly suppressed the exports of mechanical and electrical products, and intra-industry trade not only significantly promoted exports but also alleviated the inhibitory effect of EPU on firms' exports.

Second are measurement error biases, which mainly include the measurement error of EPU and the intra-industry trade index. Considering that the original data for the EPU index are at the month level, different calculation methods used in the conversion into annual data may lead to differences in EPU. Since we applied the simple arithmetic mean method for the EPU index, we used a weighted arithmetic mean method instead to recalculate the EPU index for a robustness check. The regression results are reported in Columns (3)–(4) in Table 3, where the key interaction term of intra-industry trade and EPU is not included in Column (3)

**Table 3. Robustness checks.**

| Variable | (1) | (2) | (3) | (4) | (5) | (6) |
|---|---|---|---|---|---|---|
| | lnvoh | lnvoh | lnvoh | lnvoh | lnvoh | lnvoh |
| lnEPU | -0.0792*** | -0.0763*** | | | | |
| | (-4.1958) | (-4.0436) | | | | |
| GL | 0.0035*** | 0.0034*** | | | | |
| | (14.1117) | (14.0585) | | | | |
| lnEPU×GL | | 0.0027*** | | | | |
| | | (7.1512) | | | | |
| lnWEPU1 | | | -0.0904*** | -0.0886*** | | |
| | | | (-4.8404) | (-4.7475) | | |
| GL | | | 0.0041*** | 0.0041*** | | |
| | | | (14.6092) | (14.4408) | | |
| lnWEPU1×GL | | | | 0.0021*** | | |
| | | | | (5.1887) | | |
| lnEPU | | | | | -0.0944*** | -0.0931*** |
| | | | | | (-4.5907) | (-4.5253) |
| Bi | | | | | 0.0021*** | 0.0019*** |
| | | | | | (10.2235) | (9.5529) |
| lnEPU×Bi | | | | | | 0.0019*** |
| | | | | | | (4.5361) |
| Control variables | Yes | Yes | Yes | Yes | Yes | Yes |
| Time FE | Yes | Yes | Yes | Yes | Yes | Yes |
| Country FE | Yes | Yes | Yes | Yes | Yes | Yes |
| Product FE | Yes | Yes | Yes | Yes | Yes | Yes |
| Country-product FE | Yes | Yes | No | No | No | No |
| N | 407027 | 407027 | 408291 | 408291 | 361765 | 361765 |
| adj. R2 | 0.580 | 0.580 | 0.460 | 0.461 | 0.440 | 0.440 |

Note:

***, **, and * denote $p < 0.01$, $p < 0.05$, and $p < 0.1$, respectively. Columns (1)–(6) report the regression results obtained using the different measures.

but is included in Column (4). The results showed no substantial difference in the coefficient signs and significance of our key variables with our baseline regressions, implying the robustness of our results.

There exists a common phenomenon of "imports for exports" in China, and foreign trade shows the characteristic of "big imports and big exports" as a whole. The intra-industry trade index (GL) used in the previous regressions was unable to capture the dynamic trend of intra-industry trade patterns. To this end, we used the Brülhart index to recalculate the intra-industry trade index and to identify whether our baseline results changed while using different methods to capture intra-industry trade. The new results are shown in Columns (5)–(6) of Table 3, showing that after using a new measurement method for intra-industry trade, our baseline results remained unchanged in general.

Last, but important, we considered the reverse causality between EPU and the exports of mechanical and electrical products, as well as the reverse causality between intra-industry trade and exports. Since our explained variable is at the product level, we neglected its reverse causality with EPU, which is at the country level, and focused on the latter. In the previous analysis, we defined intra-industry trade at the HS 6-digit level to fit its definition. In this paper, 44.8% of the full sample used at the HS 6-digit industry level contained only one HS

**Table 4. Instrumental variable test.**

| Variable | (1) | (2) |
|---|---|---|
| | **lnvoh** | **lnvoh** |
| **lnEPU** | 0.0046*** | 0.0045*** |
| | (9.5453) | (9.1479) |
| **GL** | -0.0836*** | -0.1164*** |
| | (-4.0414) | (-4.8658) |
| **lnEPU×GL** | | 0.0019*** |
| | | (2.7579) |
| **Control variables** | Yes | Yes |
| **Time FE** | Yes | Yes |
| **Country FE** | Yes | Yes |
| **Product FE** | Yes | Yes |
| **First stage F- statistic** | 23844.48 | 12010.00 |
| **K-P LM χ2-statistic** | 5520.225 | 4309.067 |
| **K-P Wald F-statistic** | 2.4e+04 | 1.1e+04 |
| **Hansen J-statistic** | Equation exactly identified | |
| **N** | 337435 | 337435 |
| **R2** | 0.011 | 0.011 |
| **adj. R2** | 0.007 | 0.007 |

Note:

***, **, and * denote $p < 0.01$, $p < 0.05$, and $p < 0.1$, respectively. Columns (1)–(2) show the results with control variables lagged.

8-digit product. Therefore, the measurement of the intra-industry trade naturally determined the pronounced reverse causality that may exist between it and the exports of mechanical and electrical products. Since it was not easy to find a specific instrument variable that satisfied both exogeneity and relevance conditions, we lagged the intra-industry trade index for one period as a proxy to address the potential endogeneity problem. The one-period lag of the intra-industry trade index is correlated with the current lag and may not be affected by current exports; thus, it is a reasonable instrumental variable. The results are shown in Table 4. Through an unidentified test, we found that the p value of the Kleibergen–Paap rk LM statistic is 0, which strongly rejects the null hypothesis of nonidentification. The Kleibergen–Paap rk Wald F-statistic was much larger than the cutoff value at the 10% level of the Stock-Yogo weak instrument test, indicating no weak instrument problems. Since the number of instrument variables in this paper equaled that of the endogenous explanatory variables, we defined our equation as exactly identified. The results show that after considering endogeneity problems, EPU in destination markets still significantly suppresses the exports of mechanical and electrical products, and intra-industry trade not only significantly promotes exports but can also alleviate the inhibitory effect of TPU on firms' exports, which once again proves the robustness of our baseline results.

## 4.3. Further discussions

Since the impact of EPU may be different on ordinary exports and processing exports and because processing trade accounts for nearly half of all China's exports, we need to consider the heterogeneous effect on two such export patterns [40]. In our sample of exporters, more than 70% took processing exports before 2007, and after 2007, the rate remains 30% above.

Processing trade has the feature of "two ends abroad" and mainly takes part in international circulation, while for ordinary trade, the front end (production) is based at home, and the back end (sale) is for abroad, and its production process mainly depends on domestic circulation. Therefore, we assume that heterogeneous effects of EPU exist on ordinary and processing exports. A large body of existing trade literature suggests that the degree of participation in the production link of processing trade is shallow and less sensitive to the EPU of foreign countries; thus, EPU has less impact on processing trade [41]. However, this conclusion is still debatable. Processing trade is only engaged in low-tech workflows such as simple processing and assembly of raw materials, does not participate deeply in the production link, and the fixed export costs are relatively lower [42]. From the perspective that EPU affects fixed export costs and thus suppresses exports, processing trade with low fixed export costs is indeed less sensitive to EPU. However, processing trade relies mainly on foreign orders and belongs to the typical "order economy"; thus, the value added of exports is low at home [43, 44]. In addition, at the low end of the value chain, innovation ability, core technology, and independent brands are lacking for firms engaged in processing trade. Thus, the bargaining power is weak, and the profit space is narrow. Since foreign importers (trading partners) are more sensitive to their domestic EPU, when EPU rises, the loss of foreign orders in processing trade becomes serious, which has a greater eroding effect on the profits of processing exports. From this perspective, EPU has a greater negative effect on processing exports. Based on the analysis above, we examined the impact of EPU on the exports of mechanical and electrical products in different trade patterns. Columns (1)–(2) of Table 5 are the subsample results of ordinary exports and processing exports, showing that EPU only has a negative and statistically significant effect on processing exports, with no significant effect on ordinary exports. We further divided the full sample into two subsamples: processing with imported materials and processing with customer materials. The results are reported in Columns (3)–(4) of Table 5, showing that the negative effect of EPU on processing exports with imported materials is greater than that on processing exports with foreign-supplied materials.

Why did EPU only have a negative impact on processing exports but not ordinary exports? We assume that it is a feature of the "order economy" that makes processing trade more vulnerable; for one thing, EPU increases trade costs and reduces the profit space of processing trade; for another, the weakness of foreign demand caused by rising EPU is more likely to be reflected in the decline of orders of processing trade. Therefore, we divided the ordinary and

**Table 5. Different trade patterns: ordinary trade and processing trade.**

| Variable | (1) | (2) | (3) | (4) |
|---|---|---|---|---|
| | lnvoh_ot | lnvoh_pt | lnvoh_jlpt | lnvoh_llpt |
| lnEPU | -0.0236 | -0.1549*** | -0.1385*** | -0.1049*** |
| | (-1.1691) | (-5.6338) | (-4.8788) | (-3.3634) |
| Control variables | Yes | Yes | Yes | Yes |
| Time FE | Yes | Yes | Yes | Yes |
| Country FE | Yes | Yes | Yes | Yes |
| Product FE | Yes | Yes | Yes | Yes |
| N | 385223 | 259443 | 257947 | 186476 |
| adj. R2 | 0.438 | 0.380 | 0.360 | 0.306 |

Note:

***, **, and * denote p < 0.01, p < 0.05, and p < 0.1, respectively. Columns (1)–(2) consider the subsample of ordinary exports and processing exports. Columns (3)–(4) include the subsamples of processing with imported materials and processing with customer materials.

**Table 6. Different trade margins: Quantity margins and price margins.**

| | Subsample of ordinary trade | | Subsample of processing trade | | Full sample | |
|---|---|---|---|---|---|---|
| | Quantity margin | Price margin | Quantity margin | Price margin | Quantity margin | Price margin |
| **Variable** | (1) | (2) | (3) | (4) | (5) | (6) |
| | lnqoh_ot | lnprice_ot | lnqoh_pt | lnprice_pt | lnqoh | lnprice |
| **lnEPU** | 0.0397* | -0.0461*** | -0.3245*** | -0.0364*** | -0.0490** | -0.0381*** |
| | (1.7192) | (-4.0808) | (-8.1709) | (-2.6428) | (-2.2296) | (-3.4473) |
| **GL** | 0.0027*** | 0.0001 | 0.0048*** | -0.0001 | 0.0037*** | 0.0002 |
| | (9.2036) | (0.7522) | (11.3780) | (-0.6652) | (12.4445) | (1.3961) |
| **lnEPU×GL** | | | | | 0.0016*** | 0.0010*** |
| | | | | | (3.3797) | (4.2767) |
| **Control variables** | Yes | Yes | Yes | Yes | Yes | Yes |
| **Time FE** | Yes | Yes | Yes | Yes | Yes | Yes |
| **Country FE** | Yes | Yes | Yes | Yes | Yes | Yes |
| **Product FE** | Yes | Yes | Yes | Yes | Yes | Yes |
| **N** | 385223 | 384102 | 385223 | 258559 | 408291 | 407246 |
| **adj. R2** | 0.667 | 0.800 | 0.532 | 0.773 | 0.684 | 0.808 |

Note:

***, **, and * denote $p < 0.01$, $p < 0.05$, and $p < 0.1$, respectively. Columns (1)-(2) report the subsample of ordinary and processing exports. Column (3) considers all samples.

processing exports into quantity margins and price margins for the test. Our regression results are reported in the first four columns of Table 6. The results show that EPU has a negative effect on the price margin of both ordinary and processing exports; however, it promotes the quantity margin of ordinary exports and suppresses that of processing exports. Intra-industry trade has a positive effect on the quantity margin of both ordinary exports and processing exports but has no significant effect on the price margin, which explains why EPU has no significant effect on ordinary exports but significantly suppresses processing exports. The last two columns report the regression results of the quantity margin and price margin based on full samples after considering the interaction term, indicating that EPU has a negative and statistically significant effect on both the quantity and price margin, while intra-industry trade only has a positive effect on the quantity margin but no significant effect on the other, and the deeper the degree of intra-industry trade is, the less the export quantity margin is affected by EPU.

## 5. Conclusions

Relying on highly detailed export data from the Chinese customs database from 2000–2013, we first studied the effect of EPU on China's mechanical and electrical product exports from the perspective of intra-industry trade and then further discussed the mechanism through which the export effect of EPU varies with different trade patterns. The results showed that (1) EPU can significantly inhibit China's mechanical and electrical product exports and is mainly reflected in processing exports. (2) intra-industry trade can both benefit China's mechanical and electrical product exports and alleviate the negative impact of EPU. (3) Although EPU can significantly inhibit the price margin of both ordinary and processing exports, it can significantly improve the quantity margin of ordinary exports while inhibiting that of processing exports. Therefore, EPU has no effect on ordinary exports but has a significantly negative effect on processing exports. Please respond to specific reviewer and editor comments in the box below. To review those comments, click the View Decision Letter link.

This paper has two main limitations. The first is that, due to data availability, we cannot test the impact of recent trade friction on trade. The second is that we did not directly identify the theoretical mechanism behind the regression results.

The conclusion of this paper has important policy implications. With the deepening of the international division of labor and the rising proportion of industrial trade, the government should pay more attention to the impact of economic policy uncertainty on intra-industry trade. First, the government should prevent the risk of economic supply chain disruption and back up some key industries in the home country. Second, the government should diversify the trade markets, optimize the industrial structure of trade, rivet the global supply chain, and promote the stability of foreign trade. The government should give overall consideration to both international and domestic marketplaces and resources to open up the links between internal and external markets and accelerate the formation of a dual circulation development pattern.

The conclusion of this study also shows that processing trade, which is characterized by international procurement of intermediate products, cannot be a "talisman" for the export of mechanical and electrical products under an uncertain environment. Being at the low end of the value chain for a long time is more likely to become a "victim" under the fluctuation of the export environment. General trade is not sensitive to the uncertainty of economic policy because its production links are rooted in the domestic market. Therefore, on the one hand, we should actively make good use of the domestic intermediate products market for domestic circulation, encourage processing trade enterprises to increase the domestic procurement rate of intermediate products to improve the domestic value chain and change the "order economy" into a "demand economy". This is of positive significance for stabilizing foreign trade under the increasing uncertainty of the international environment.

The existing literature has not fully considered the link between the characteristics of intra-industry trade and policy uncertainty. Moreover, the existing literature has paid little attention to the impact of specific industry trade. Future research can discuss the impact of uncertainty on the trade of other industries.

## Author Contributions

**Conceptualization:** Dajun Liu, Xiugang Zhu.

**Data curation:** Xiugang Zhu.

**Formal analysis:** Xiugang Zhu.

**Funding acquisition:** Xiugang Zhu.

**Investigation:** Xiugang Zhu.

**Methodology:** Xiugang Zhu, Huiru Yu.

**Project administration:** Xiugang Zhu.

**Resources:** Xiugang Zhu.

**Software:** Xiugang Zhu, Huiru Yu.

**Supervision:** Xiugang Zhu.

**Validation:** Xiugang Zhu.

**Visualization:** Xiugang Zhu.

**Writing – original draft:** Xiugang Zhu.

**Writing – review & editing:** Dajun Liu, Xiugang Zhu, Huiru Yu.

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
