## [Decision Letter · Decision Letter 0]

20 Sep 2022

PONE-D-22-06611Economic Policy Uncertainty, Intra-industry Trade and China’s Mechanical and Electrical Product ExportsPLOS ONE

Dear Dr. Zhu,

Thank you for submitting your manuscript to PLOS ONE. After careful consideration, we feel that it has merit but does not fully meet PLOS ONE’s publication criteria as it currently stands. Therefore, we invite you to submit a revised version of the manuscript that addresses the points raised during the review process.

We look forward to receiving your revised manuscript.

Kind regards,

Atif Jahanger, Ph.D

Academic Editor

PLOS ONE

Journal Requirements:

2. PLOS requires an ORCID iD for the corresponding author in Editorial Manager on papers submitted after December 6th, 2016. Please ensure that you have an ORCID iD and that it is validated in Editorial Manager. To do this, go to ‘Update my Information’ (in the upper left-hand corner of the main menu), and click on the Fetch/Validate link next to the ORCID field. This will take you to the ORCID site and allow you to create a new iD or authenticate a pre-existing iD in Editorial Manager. Please see the following video for instructions on linking an ORCID iD to your Editorial Manager account: https://www.youtube.com/watch?v=_xcclfuvtxQ.

Reviewers' comments:

Reviewer's Responses to Questions

**Comments to the Author**

1. Is the manuscript technically sound, and do the data support the conclusions?

Reviewer #1: Partly

Reviewer #2: Yes

2. Has the statistical analysis been performed appropriately and rigorously? 

Reviewer #1: No

Reviewer #2: Yes

3. Have the authors made all data underlying the findings in their manuscript fully available?

Reviewer #1: No

Reviewer #2: Yes

4. Is the manuscript presented in an intelligible fashion and written in standard English?

Reviewer #1: Yes

Reviewer #2: Yes

5. Review Comments to the Author

Reviewer #1: I am writing about the manuscript (PONE-D-22-06611) entitled “Economic Policy Uncertainty, Intra-industry Trade, and China’s Mechanical and Electrical Product Exports”.

Thank you for the opportunity to review this paper. It is interesting and provides some good insight into the existing literature. I recommend for the paper substantial modifications and refinements of the present version. My comments are as follows:

(1) The English language needs more work. There are many grammatical and typo mistakes in this manuscript. The paper needs to be edited by a native English speaker.

(2) The major defect of this study is the debate or argument is not clearly stated in the introduction session. Hence, the contribution is weak in this manuscript. The authors have presented a lot of data and figures in the introduction section. I would suggest the authors enhance your theoretical discussion and arrive at your theoretical argument.

(3) The motivations behind the selection of Economic Policy Uncertainty, Intra-industry Trade, and China’s Mechanical and Electrical Product Exports countries are not clear.

(4) A graphical presentation of study hypothesis will make it more readable and attract the audience.

(5) Introduction, literature review, and empirical results and discussion section should be critically evaluated by the authors. I recommend improving this section by critically analyzing the previous studies.

(6) Literature gap is missing at the end of the literature review section. In this regard, the authors should mention the drawbacks and limitations of the previous studies and also mention that how to address these issues solved in this study accordingly.

(7) The references of applied data are missing in the main text.

(8) More explanations and interpretations must be added for the results. In this regard, it is suggested to compare the results of the present research with some similar studies which is done before and more explanations and interpretations must be added for the Results, and discussion, which are not enough.

(9) The authors should present the main findings in graphical form. It will increase the brevity and more readerships and attract more audience.

(10) The conclusion and policy recommendations are not well written. Authors should add more to this section, especially in the aspect of policy framing and implementation.

(11) It would be appropriate to indicate future research directions at the end of the conclusion section just before references.

Reviewer #2: The authors have attempted to test the effect of economic policy uncertainty (EPU) and domestic intra-industry trade on China’s mechanical and electrical product exports to 23 trading partners. The overall work is good. However, I feel that there is still some room to improve the paper before it can be published. My suggestions are attached in a separate file.

6. PLOS authors have the option to publish the peer review history of their article (what does this mean?). If published, this will include your full peer review and any attached files.

Reviewer #1: No

Reviewer #2: **Yes: **Mohammad Razib Hossain

---

## [Author Response · Author response to Decision Letter 0]

6 Dec 2022

We have revised the article as requested by the editor, see Response to Reviewer Report for more details Response to Reviewer Report.

---

## [Decision Letter · Decision Letter 1]

8 Jan 2023

PONE-D-22-06611R1Economic Policy Uncertainty, Intra-industry Trade and China’s Mechanical and Electrical Product ExportsPLOS ONE

Dear Dr. Zhu,

Thank you for submitting your manuscript to PLOS ONE. After careful consideration, we feel that it has merit but does not fully meet PLOS ONE’s publication criteria as it currently stands. Therefore, we invite you to submit a revised version of the manuscript that addresses the points raised during the review process.

We look forward to receiving your revised manuscript.

Kind regards,

Atif Jahanger, Ph.D

Academic Editor

PLOS ONE

Reviewers' comments:

Reviewer's Responses to Questions

**Comments to the Author**

1. If the authors have adequately addressed your comments raised in a previous round of review and you feel that this manuscript is now acceptable for publication, you may indicate that here to bypass the “Comments to the Author” section, enter your conflict of interest statement in the “Confidential to Editor” section, and submit your "Accept" recommendation.

Reviewer #1: All comments have been addressed

Reviewer #2: (No Response)

2. Is the manuscript technically sound, and do the data support the conclusions?

Reviewer #1: Yes

Reviewer #2: Partly

3. Has the statistical analysis been performed appropriately and rigorously? 

Reviewer #1: Yes

Reviewer #2: Yes

4. Have the authors made all data underlying the findings in their manuscript fully available?

Reviewer #1: Yes

Reviewer #2: Yes

5. Is the manuscript presented in an intelligible fashion and written in standard English?

Reviewer #1: Yes

Reviewer #2: No

6. Review Comments to the Author

Reviewer #1: (No Response)

Reviewer #2: Please refer to the attached PDF file for a list of comments. The manuscript has been naively crafted and lacks professionalism. All the comments need to be addressed to be able to get published.

7. PLOS authors have the option to publish the peer review history of their article (what does this mean?). If published, this will include your full peer review and any attached files.

Reviewer #1: No

Reviewer #2: **Yes: **Mohammad Razib Hossain

---

## [Author Response · Author response to Decision Letter 1]

13 Apr 2023

Response to Reviewer Report on PONE-D-22-06611

“Economic Policy Uncertainty, Intra-industry Trade, and China’s Mechanical and Electrical Product Exports”

We would like to thank you for your valuable comments and suggestions. Following these comments, we have once again revised the manuscript. We believe that the quality of our manuscript has been greatly improved. The detailed responses are shown below. First, you point out that there are many grammatical errors in the paper, we have asked professionals to revise them; second, we have added instructions on the importance of the research content; and finally, we have made detailed modifications for your other suggestions. Furthermore, in our resubmitted draft, the revised section is highlighted in red.

Response to Reviewer

The purpose of this paper is to empirically investigate the impact of economic policy uncertainty and domestic intra-industry trade on China's exports of electromechanical products to its 23 trading partners. Then the role of intra-industry trade on the regulation of EPU and export is further discussed. The authors point out that EPU has a significant inhibitory effect on the export of electromechanical products; instead, internal industrial trade can both significantly promote the export and alleviate the inhibitory effect of EPU. Moreover, due to the differences in trade patterns, EPU significantly inhibits processing and exports, but it has no impact on ordinary exports. The above results proved to be robust by endogenous problems. 

Overall, this paper finds that the relationship between trade policy and exports should be valued, and I will continue to extend my attention and provide some relevant suggestions below.

Thank you for the insightful comments. We have substantially revised our manuscript. Please see the revised paper and our responses below for more details.

Comment:

1. In the abstract, the authors have added two lines in the beginning. However, these do not justify why the work is important and it requires attention from the wider scholarly community. I recommend that the authors incorporate the importance of this work (why the author feels that this work is important). 

Response: We concur with your comments. We think that the extant related literature

rarely analyzes the nexus between uncertainty and trade on a specific industry, and our research focuses on the mechanical and electrical industry. Please see the revised paper for more details (section 1, page 1).

2. “The study of this paper indicates that in the context of increasing uncertainty, our findings could have far-reaching policy implications for China to build a new development pattern of domestic and international dual circulation”. Please write “The results of this paper indicate that ……

Response: On the basis of your suggestions, our revised results are presented as follows (section 1, page 1):

“The results of this paper indicate that in the context of increasing uncertainty, our findings could have far-reaching policy implications for China to build a new development pattern of domestic and international dual circulation.”

3. There are many scattered typos visible in the naked eye. Still the authors have failed to address them. This indicates lack of professionalism. On page 1, line 30, I see “0.” attached to the citation (i.e., Handley, 20140.). Why is that? Similar cases are also seen in lines 36, 52 …. Section 2.1, 2.2, 22.3 in the LR as well. I see them all through the manuscript. Please remove them all. 

Response: Thank you for your comments. Listen to your suggestions, we rechecked the entire paper and revised. Please see the revised paper for more details.

4. On page 1, line 41, it should be China’s. Please read full manuscript with attention and care to fix all these typos. It is not possible to point out all different typos, given that it is your responsibility to identify and fix them. 

Response:Thanks for your suggestions. Based on the suggestions you gave, we have modified this spelling error (section 1, page 2).

5. Use a different color in figure 1, exclusively for the dotted line. Its not very clear at this point. 

Response: Thank you for your comments. We modified Fig 1, more details are shown below.

Figure 1 The proportion of the export volume of mechanical and electrical products

6. In the footnote 1, on page 2, what is 82-91?? Please clarify. 

Response: Thank you for your comments. We have added the meaning of this number in this paper. More details are shown below (section 1, page 3).

“The mechanical and electrical products mentioned in the article refer to the products whose HS code range from 82 to 91 in the customs data.”

7. Line 51 is incorrectly written. Please fix the grammatical errors. 

Response: Thanks for your suggestions. We have fixed these grammatical errors, more details are shown below (section 1, page 2).

“Extensive literature on the relationship between EPU and trade (e.g., Andrew,2019 [6], Gervais, 2018 [7]) does not consider the correlation between the characteristics of intraindustry trade and EPU.”

8. There are so too much smaller paragraphs in the intro. Merge them and produce reasonably sized paragraphs. 

Response: Thank you for your suggestions. Listen to your suggestions, we have integrated the passages of this paper. More details are shown below (section 1, page 3).

9. The novelty section should be written in a concrete structure. Currently, it is scattered. Place the novelty section in the end of the introduction. Elaborate the novelties within a point-by-point framework (i.e., firstly, secondly….).

Response: Thank you for your suggestions. Listen to your suggestions, we have made the modifications. More details are shown below (section 1, page 4).

“The marginal contribution of this paper is mainly reflected in three aspects. First, the nexus between uncertainty and exports is analyzed from the perspective of a specific industry. The extant related literature mainly analyzes the nexus between uncertainty and trade across different industries and focuses less on a specific industry. Second, we empirically test the moderation effect of intraindustry trade and trade patterns on the relationship between uncertainty and exports based on highly disaggregated trade data. Third, the conclusions have some crucial policy implications. Further involvement in intraindustry trade and diversification of trade patterns can help firms alleviate the impact of external shocks due to policy uncertainty.”

10. I do not understand what the authors intend to say in line 64, “As shown in the figure 1……..”. It seems to be falling apart. 

Response: Thank you for your comments. We have reillustrated Fig 1. More details are shown below (section 1, page 2).

“ As shown in Fig. 1, the proportion of exports of electrical and mechanical products to total exports has been above 50% since 2003.”

11. In the intro, please try to establish a robust link between the core of the problem and how you can address the issues empirically. At this moment it is missing. This comment was given in the initial round. Please take care of this comment. I did not ask the authors to provide justification about why they used the specific empirical model. Rather, I asked them to discuss the core problem based on which this paper has been written and the ways to tackle the core problem. 

Response: Thank you for your suggestions. Listen to your suggestions, we have added to this content in the introduction. More details are shown below (section 1, page 3).

“The core issue of this paper is to identify the impact of EPU on mechanical and electrical products and the moderating role of intraindustry trade. To make the empirical results more credible, we addressed the endogeneity problem in three aspects. First, the omitted variable generally causes an error term correlated with other explanatory variables, which leads to a violation of the exogenous assumption. To solve this problem, we further control the product-country fixed effects according to the baseline regression. Second, the presence of measurement error biases causes explanatory variables to be correlated with perturbation terms, resulting in an endogeneity problem, so we use the weighted arithmetic mean method to recalculate the EPU index. Finally, reverse causality also causes a potential endogeneity problem, and we lag the intraindustry trade index for one period as a proxy to address this problem. After alleviating the three endogeneity problems, the conclusions remain robust.”

12. Start line 89 as, “The remaining of the manuscript is structured as follows:……” 

Response: Thank you for your suggestions. Listen to your suggestions, we have made the modifications. More details are shown below (section 1, page 4).

“The remainder of the paper is structured as follows. We show the relevant literature and theoretical hypotheses in Section 2. Based on the related literature, we propose three theoretical hypotheses to be tested. We present the estimation strategy in Section 3, mainly by establishing the regression models. Section 4 shows the regression results, including the benchmark regression results and a series of robustness tests. Section 5 presents the conclusion and policy implications.”

13. Incorporate a separate paragraph in the end of the LR section and write the overall gaps in the literature. 

Response: Thank you for your suggestions. Listen to your suggestions, we have made relevant supplements. More details are shown below (section 1, page 2).

“Extensive literature on the relationship between EPU and trade (e.g., Andrew,2019 [6], Gervais, 2018 [7]) does not consider the correlation between the characteristics of intraindustry trade and EPU. The impact of intraindustry trade may be more sensitive to EPU. At the same time, the literature focuses less on the impact of a certain industry and ignores the differences in the impact of uncertainty on the trade of different industries. This paper focuses on the impact of EPU on exports of the mechanical and electrical products industry and conducts an empirical study using customs data. We find that EPU has a significant inhibition effect on mechanical and electrical product exports; conversely, intraindustry trade can both significantly promote exports and alleviate the inhibition effect of EPU. In addition, the export impact of EPU varied with different trade patterns. It can significantly inhibit processing exports, while it has no effect on ordinary exports.”

14. There is no discussion on the theoretical mechanism and no discussion about figure 2 

Response: On the basis of your suggestions, we have added an explanation of the theoretical mechanism and Fig 2. More details are shown below (section 1, page 6).

Figure 2 Theoretical Mechanism

“Fig. 2 is the mechanism diagram depicted in this paper, where H1 illustrates the inhibitory effect of EPU on the export of mechanical and electrical products, H2 indicates the promotion effect of intraindustry trade on the export of mechanical and electrical products, and H3 indicates that it can alleviate the inhibitory effect of EPU on the export of mechanical and electrical products.”

15. In table 1, highlight and bold the key points (such as explained variable, key explanatory variables, control variable) 

Response: On the basis of your suggestions, we have modified the tables in this paper. More details are shown below.

16. On page 10, line 342, Kee and Tang, 20160, it should be 2016. 

Response: On the basis of your suggestions, we have fixed this error in this paper. More details are shown below (section 1, page 12).

17. The authors have incorporated table notes under each of them. However, there are many visible spelling mistakes that the authors did not care to fix. They just copied and pasted same lines again and again. This is unprofessional and shameful. This is a revised version and the authors should put extra care in fixing the issues. 

Response: On the basis of your suggestions, we have modified the table notes in this paper. More details are shown below.

18. The English language needs substantial work and I recommend that the authors visit a professional proofreader to get the job done. 

Response: On the basis of your suggestions, we have invited professionals to proofread the paper.

19. Please include the limitations of the present study and the direction for any future studies in the end of the conclusion. NOT in the middle.

Response: On the basis of your suggestions, we have supplemented the relevant content in the end of this paper. More details are shown below (section 1, page 14).

“The existing literature has not fully considered the link between the characteristics of intraindustry trade and policy uncertainty. Moreover, the existing literature has paid little attention to the impact of specific industry trade. Future research can discuss the impact of uncertainty on the trade of other industries.”

---

## [Decision Letter · Decision Letter 2]

8 Jun 2023

PONE-D-22-06611R2Economic Policy Uncertainty, Intraindustry Trade and China’s Mechanical and Electrical Product ExportsPLOS ONE

Dear Dr. Zhu,

Thank you for submitting your manuscript to PLOS ONE. After careful consideration, we feel that it has merit but does not fully meet PLOS ONE’s publication criteria as it currently stands. Therefore, we invite you to submit a revised version of the manuscript that addresses the points raised during the review process.

We look forward to receiving your revised manuscript.

Kind regards,

Atif Jahanger, Ph.D

Academic Editor

PLOS ONE

Journal Requirements:

Reviewers' comments:

Reviewer's Responses to Questions

**Comments to the Author**

1. If the authors have adequately addressed your comments raised in a previous round of review and you feel that this manuscript is now acceptable for publication, you may indicate that here to bypass the “Comments to the Author” section, enter your conflict of interest statement in the “Confidential to Editor” section, and submit your "Accept" recommendation.

Reviewer #1: All comments have been addressed

Reviewer #2: (No Response)

2. Is the manuscript technically sound, and do the data support the conclusions?

Reviewer #1: Partly

Reviewer #2: Yes

3. Has the statistical analysis been performed appropriately and rigorously? 

Reviewer #1: Yes

Reviewer #2: Yes

4. Have the authors made all data underlying the findings in their manuscript fully available?

Reviewer #1: No

Reviewer #2: Yes

5. Is the manuscript presented in an intelligible fashion and written in standard English?

Reviewer #1: Yes

Reviewer #2: Yes

6. Review Comments to the Author

Reviewer #1: The authors have addressed my all comments well. Therefore, this study can be accepted in this journal.

Reviewer #2: The paper at this point looks better than the earlier version. However, the following changes should be added before the acceptance.

1. Fix the following: change "intraindustry" to "intra-industry". This error is prevalent all throughout the manuscript from title to the conclusion section. Proofread the paper and fix all.

2. Add a motivation section in the introduction to discuss what motivates you to write this paper.

3. The theoretical framework is missing. Please add a theoretical framework section in the beginning of the methodology and discuss the theories of the relevant research questions and how they are associated with your research hypotheses. "Theoretical mechanism" does not convey the theoretical underpinnings.

4. Please make sure that all titles and sub-titles are written in sentence case format (i.e., section 4.1 is different). Or at least they should follow the same pattern.

5. Rename section 5 as conclusions and policy implications. You should have at least three paragraphs here. First, convey what you did and convey the main results. Second, you should have the policy implications in a point-by-point framework. And third, you should have the limitations and prospect for future studies.

7. PLOS authors have the option to publish the peer review history of their article (what does this mean?). If published, this will include your full peer review and any attached files.

Reviewer #1: No

Reviewer #2: No

---

## [Author Response · Author response to Decision Letter 2]

23 Jul 2023

Response to Reviewer Report on PONE-D-22-06611

“Economic Policy Uncertainty, Intra-industry Trade, and China’s Mechanical and Electrical Product Exports”

We would like to thank you for your valuable comments and suggestions. Following these comments, we have thoroughly revised the manuscript. We believe that the quality of our manuscript has been greatly improved. The detailed responses are shown below. First, your comments and suggestions are errors in grammar and text that we have modified. Secondly, in the abstract and introduction part of your comments and suggestions are not rich and detailed enough, we have added the main conclusions of the article and the empirical test content, expanded the content of the theoretical part. Finally, we need to be improved in what your comments and suggestions are about the conclusions and policy suggestions, and we have added new content. Furthermore, in our resubmitted draft, the revised section is highlighted in blue.

Response to Reviewer

The purpose of this paper is to empirically investigate the impact of economic policy uncertainty and domestic intra-industry trade on China's exports of electromechanical products to its 23 trading partners. Then the role of intra-industry trade on the regulation of EPU and export is further discussed. The authors point out that EPU has a significant inhibitory effect on the export of electromechanical products; instead, internal industrial trade can both significantly promote the export and alleviate the inhibitory effect of EPU. Moreover, due to the differences in trade patterns, EPU significantly inhibits processing and exports, but it has no impact on ordinary exports. The above results proved to be robust by endogenous problems. 

Overall, this paper finds that the relationship between trade policy and exports should be valued, and I will continue to extend my attention and provide some relevant suggestions below.

Thank you for the insightful comments. We have substantially revised our manuscript. Please see the revised paper and our responses below for more details.

Comment :

（1）Fix the following: change "intraindustry" to "intra-industry". This error is prevalent all throughout the manuscript from title to the conclusion section. Proofread the paper and fix all.

Response:On the basis of your suggestions,I have changed the full paper of "intraindustry" to "intra-industry".Please see the revised paper for more details.

（2）Add a motivation section in the introduction to discuss what motivates you to write this paper.

Response:We concur with your comments.Below is the specific content of the changes provided.

“ Extensive literature on the relationship between EPU and trade (e.g., Andrew,2019 [6], Gervais, 2018 [7]) does not consider the correlation between the characteristics of intra-industry trade and EPU. The impact of intra-industry trade may be more sensitive to EPU. At the same time, the literature focuses less on the impact of a certain industry and ignores the differences in the impact of uncertainty on the trade of different industries. In this context, the paper focuses on the electromechanical products industry to address this research gap. This paper aims to clarify the impact mechanism of trade uncertainty on export trade within specific industries. On one hand, it will facilitate our understanding of the changing trends in foreign trade under the backdrop of uncertainty from a global value chain perspective. On the other hand, it will be beneficial in providing relevant policy insights for emerging market countries like China to address economic policy uncertainties.This paper focuses on the impact of EPU on exports of the mechanical and electrical products industry and conducts an empirical study using customs data. We find that EPU has a significant inhibition effect on mechanical and electrical product exports; conversely, intra-industry trade can both significantly promote exports and alleviate the inhibition effect of EPU. In addition, the export impact of EPU varied with different trade patterns. It can significantly inhibit processing exports, while it has no effect on ordinary exports.”

（3）The theoretical framework is missing. Please add a theoretical framework section in the beginning of the methodology and discuss the theories of the relevant research questions and how they are associated with your research hypotheses. "Theoretical mechanism" does not convey the theoretical underpinnings.

Response:We concur with your comments.Below is the specific content of the changes provided.

“The classic theories relevant to this paper are primarily the theory of monopolistic competition in international trade (Krugman 1980[32];Melitz 2003[33] ). This theory is mainly based on empirical facts from developed countries and explains the role of internal economies of scale and firm heterogeneity in international trade patterns. Subsequently, some scholars incorporated uncertainty into the analytical framework and extended the models of heterogeneous firm trade from various perspectives (Handley and Limão, 2022 [1] ,2017[5] ).The theoretical framework of this article primarily highlights the significant role of economic policy uncertainty in global trade, particularly when it comes to specific industries, where uncertainty can sharply increase. Within this theoretical framework, the article explores the relevant research questions, and the theoretical mechanism is illustrated in Fig. 2.”

（4）Please make sure that all titles and sub-titles are written in sentence case format (i.e., section 4.1 is different). Or at least they should follow the same pattern.

Response: Thank you for your comments.I have made sure that all titles and sub-titles in sentence case format.Please see the revised paper for more details.

（5）Rename section 5 as conclusions and policy implications. You should have at least three paragraphs here. First, convey what you did and convey the main results. Second, you should have the policy implications in a point-by-point framework. And third, you should have the limitations and prospect for future studies.

Response:Thank you for your comments.I have already renamed Section 5 as Conclusions and Policy Implications, and made the necessary modifications to the paragraph content as you requested.Please see the revised paper for more details.

（6）PLOS authors have the option to publish the peer review history of their article (what does this mean?). If published, this will include your full peer review and any attached files.

Response:Thank you for your comments.

---

## [Decision Letter · Decision Letter 3]

18 Aug 2023

Economic Policy Uncertainty, Intraindustry Trade and China’s Mechanical and Electrical Product Exports

PONE-D-22-06611R3

Dear Dr. Zhu,

We’re pleased to inform you that your manuscript has been judged scientifically suitable for publication and will be formally accepted for publication once it meets all outstanding technical requirements.

Kind regards,

Atif Jahanger, Ph.D

Academic Editor

PLOS ONE

Additional Editor Comments (optional):

Reviewers' comments:

Reviewer's Responses to Questions

**Comments to the Author**

1. If the authors have adequately addressed your comments raised in a previous round of review and you feel that this manuscript is now acceptable for publication, you may indicate that here to bypass the “Comments to the Author” section, enter your conflict of interest statement in the “Confidential to Editor” section, and submit your "Accept" recommendation.

Reviewer #2: All comments have been addressed

2. Is the manuscript technically sound, and do the data support the conclusions?

Reviewer #2: Yes

3. Has the statistical analysis been performed appropriately and rigorously? 

Reviewer #2: Yes

4. Have the authors made all data underlying the findings in their manuscript fully available?

Reviewer #2: Yes

5. Is the manuscript presented in an intelligible fashion and written in standard English?

Reviewer #2: Yes

6. Review Comments to the Author

Reviewer #2: All good. Please put the limitations in the end of your conclusion. ................................No more comments

7. PLOS authors have the option to publish the peer review history of their article (what does this mean?). If published, this will include your full peer review and any attached files.

Reviewer #2: No

---

## [Editor Report · Acceptance letter]

19 Sep 2023

PONE-D-22-06611R3 

Economic Policy Uncertainty, Intra-industry Trade, and China’s Mechanical and Electrical Product Exports 

Dear Dr. Zhu:

I'm pleased to inform you that your manuscript has been deemed suitable for publication in PLOS ONE. Congratulations! Your manuscript is now with our production department. 

Kind regards, 

on behalf of

Dr. Atif Jahanger 

Academic Editor

PLOS ONE